# Determination of Trimethylamine *N*-oxide and Betaine in Serum and Food by Targeted Metabonomics

**DOI:** 10.3390/molecules26051334

**Published:** 2021-03-02

**Authors:** Mingshuai He, Heshui Yu, Peng Lei, Shengjie Huang, Juanning Ren, Wenjing Fan, Lifeng Han, Haiyang Yu, Yuefei Wang, Ming Ren, Miaomiao Jiang

**Affiliations:** 1State Key Laboratory of Component-Based Chinese Medicine, Tianjin University of Traditional Chinese Medicine, Tianjin 301617, China; h1115049914@163.com (M.H.); hs_yu08@163.com (H.Y.); leipengcn@163.com (P.L.); huang962021@sina.com (S.H.); 18298419332@163.com (J.R.); fwj4239328@163.com (W.F.); hanlifeng_1@163.com (L.H.); yuhaiyang19830116@hotmail.com (H.Y.); 13516185421@139.com (Y.W.); 2Department of Pharmacy, Institute of Traditional Chinese Medicine, Tianjin University of Traditional Chinese Medicine, Tianjin 301617, China; 3Institute of Traditional Chinese Medicine, Tianjin University of Traditional Chinese Medicine, 10 Poyanghu Road, West Area, Tuanbo New Town, Jinghai District, Tianjin 301617, China

**Keywords:** TMAO, NMR, quantification, cation exchange solid-phase extraction, gender differences

## Abstract

Trimethylamine *N*-oxide (TMAO), as a gut-derived metabolite, has been found to be associated with enhanced risk for atherosclerosis and cardiovascular disease. We presented a method for targeted profiling of TMAO and betaine in serum and food samples based on a combination of one-step sample pretreatment and proton nuclear magnetic resonance spectroscopy. The key step included a processing of sample preparation using a selective solid-phase extraction column for retention of basic metabolites. Proton signals at *δ* 3.29 and *δ* 3.28 were employed to quantify TMAO and betaine, respectively. The developed method was examined with acceptable linear relationship, precision, stability, repeatability, and accuracy. It was successfully applied to detect serum levels of TMAO and betaine in TMAO-fed mice and high-fructose-fed rats and also used to determine the contents of TMAO and betaine in several kinds of food, such as fish, pork, milk, and egg yolk.

## 1. Introduction

Trimethylamine *N*-oxide (TMAO) is an oxidized product of trimethylamine (TMA), which is generated in the gut from betaine, butyrobetaine, carnitine, choline, and other choline-containing compounds [1,2]. Eggs, milk, liver, red meat, poultry, shell fish, and fish are recognized as the major dietary sources for choline and betaine [3]. The initial catabolism of choline and other TMA-containing species (e.g., betaine) by intestinal microbes form the gas TMA [4], which is efficiently absorbed and rapidly metabolized by at least one member of the hepatic flavin monooxygenase (FMO) family to generate TMAO [5,6]. The blood level of TMAO is influenced by a number of factors, such as diets, intestinal flora, drug regulation, and liver FMO activity [7]. A high level of TMAO exhibits clinical relevance with several chronic diseases, including inflammation [8,9], atherosclerosis [10], cardiovascular disease [11], metabolic syndrome [12], and neurological disorders [13,14]. In addition, dietary betaine has been found in association with incident of coronary heart disease in African-American participants [15]. Therefore, it is necessary to establish an accurate and efficient method for TMAO and betaine determination.

Some methods have been reported to quantify TMAO and betaine in biological and food samples but with several limitations. For example, determination of TMAO and betaine in plasma based on liquid chromatography tandem mass spectrometry (LC–MS) [16,17] requires isotopic labeled standards (trimethylamine *N*-oxide-*d*_9_ and betaine-*d*_11_), which are prohibitively expensive. Proton nuclear magnetic resonance (^1^H NMR) technology has some advantages when determining metabolites in biological samples. For a given proton, the signal strength or area is directly proportional to the number of protons producing the signal, regardless of the nuclear chemistry. Moreover, the content of a compound in a mixture can be calculated without its reference. The ^1^H NMR method is simple, rapid and specific. However, biological and food samples usually contain various metabolites. The proton signals of glucose and taurine in ^1^H NMR spectrum overlap with those of TMAO and betaine in a very narrow range of *δ* 3.25–3.30 [18]. Although the signal of TMAO can be separated from glucose by adjusting pH value [18], the interference from taurine signals still exists. Therefore, it is necessary to optimize sample processing condition before NMR analysis to quantify TMAO and betaine simultaneously. One of the key points of this study was the preparation of biological and food samples.

Herein, we developed a convenient, fast, and effective method to determine TMAO and betaine based on ^1^H NMR technology. Only ordinary standards were required, not the labeled ones. All the operations were performed at room temperature. A selective solid-phase extraction (SPE) column was used to separate TMAO and betaine with glucose, taurine, and other neutral or acidic metabolites whose signals were probably overlapped with those of TMAO and betaine. After SPE treatment, the signals of betaine and TMAO were obtained and could be employed to calculate peak areas.

## 2. Results

### 2.1. Sample Preparation for ^1^H NMR Measurement

As shown in Figure 1A, ^1^H NMR spectrum of biological sample is usually complicated. The proton signals of methyl groups in TMAO and betaine are overlapped with the signals of methylene group in the taurine and methenyl group in glucose, especially in the range from *δ* 3.20 to 3.40 ppm (Figure 1B). The solid-phase extraction (SPE) method was employed to extract TMAO and betaine before NMR analysis. We tested three kinds of SPE columns, including an Oasis MCX column (Waters Corporation, Milford, MA, USA) with strong cation-exchange sorbent, an Oasis WCX column (Waters Corporation, Milford, MA, USA) with weak cation-exchange sorbent, and a HyperSep^TM^ Retain CX column (Thermo Fisher Scientific Inc., Waltham, MA, USA) with cation-exchange sorbent. Compared with the original standard solution (Figure 1C,D), the HyperSep^TM^ Retain CX column showed an acceptable effect on extracting TMAO and betaine, and the disturbance signals of taurine and glucose were removed (Figure 1E,F). The result indicated HyperSep^TM^ Retain CX column had appropriate cation-exchange sorbent to separate alkaline metabolites (e.g.,TMAO, betaine) from neutral and acidic metabolites (e.g., glucose and taurine).

### 2.2. Quantitative ^1^H NMR Analysis and Method Validation

The resonance signals at *δ* 3.29 and *δ* 3.28 ppm were assigned to TMAO and betaine, respectively, according to standard references and heteronuclear multiple-bond correlation spectrum. Spin-lattice relaxation time (*T*_1_) values of TMAO and betaine were accurately measured to be 2.64 ± 0.01 s and 2.00 ± 0.01 s, respectively. For a quantitative purpose, the time of relaxation delay (D1) was accordingly set to 13 s and about five times the *T*_1_ values. The contents of TMAO and betaine were calculated using calibration curves. Method validation was performed based on previous literature [19].

In order to determine the linearity of ^1^H NMR method, five standard solutions of TMAO and betaine were prepared and tested in triplicate. The calibration curve was constructed by plotting the ratio between the peak areas of TMAO/betaine (*x*) and the internal standard versus TMAO/betaine concentration (*y*, µM) (Figure 2). For TMAO with the concentrations from 5.21 to 52.1 mM, the linearity regression yielded a correlation coefficient of 1.00 and a regression equation of *y* = 30.66 *x* − 0.3023. The LOD and LOQ values were determined as 1.00 µM and 3.02 µM, respectively. For betaine with the concentrations from 20.60 to 206.0 mM, the linearity regression yielded a correlation coefficient of 1.00 and a regression equation of *y* = 29.38 *x* − 0.4606. The LOD and LOQ values were determined to be 1.52 and 4.60 µM, respectively.

The precision of ^1^H NMR method was evaluated by six replicate measurements of the same sample, and the relative standard deviation (RSD) values of precision for the contents of TMAO and betaine were found to be 0.85% and 0.93%, respectively. The repeatability was evaluated by analyzing six different sample solutions independently prepared from the same sample. The RSD values of the contents of TMAO and betaine were found to be 3.06% and 2.76%, respectively. The stability was assessed by analyzing the same sample solution at an interval of every 2 h. The analytes were found to be stable during the tested period at ambient temperature. The RSD values for the contents of TMAO and betaine were 2.79% and 2.81%, respectively (Table 1).

The accuracy of ^1^H NMR method was determined by spiked recovery test for comparing the calculated value of added reference material and the test value. Three portions of 12 µmol TMAO and 48 µmol betaine were weighed in parallel, and 6, 12, and 18 µmol TMAO and 24, 48, and 72 µmol betaine were subsequently added in the solutions as shown in Table 2 to prepare samples with three different concentrations of TMAO and betaine. All the samples were subjected to SPE column and eluted in order. The obtained alkaline elution was subjected to NMR analysis. The actual contents were calculated according to equations of linear regression. The average recoveries were 98.48% and 98.84% with RSD values of 0.23% and 0.36% for TMAO and betaine, respectively (Table 2). Moreover, we also performed the recovery test on serum sample. Accurately weighed TMAO and betaine were added into 200 µL serum, and then subjected to SPE column for gradient elution. The obtained alkaline fraction was analyzed and quantified. The average recoveries were calculated as 98.47% and 97.95% with RSD values of 0.51% and 0.65% for TMAO and betaine, respectively (Table 3). These results indicated a good linearity, precision, repeatability, stability, and accuracy of the established method.

### 2.3. Applications to Serum and Food Samples

Using the developed ^1^H NMR method, the contents of TMAO and betaine in serum and food samples were measured. Mouse serum samples were subjected to HyperSep^TM^ Retain CX column for retention of TMAO and betaine (Figure 3A–D). The quantitative result (Figure 3E) showed TMAO level in serum changed significantly after TMAO feeding in both male and female mice (*p* < 0.05). Moreover, TMAO levels in serum of female mouse were higher than those of male ones before or after TMAO feeding. It has been reported different expression levels of FMO3 between genders (in females 1000-fold higher than in males) [1]. Increased expression of FMO3 catalyzes more TMA to produce TMAO.

Rat serum samples were preconditioned through the SPE column to obtain TMAO and betaine fraction (Figure 4A,B). ^1^H NMR analysis result (Figure 4C) showed that TMAO level in serum significantly increased (*p* < 0.05) in fructose-fed group (M) compared with that of control group (C), indicating a high-fructose diet induced enhancing TMAO level in serum.

In addition, TMAO and betaine in egg yolk, pork, fish meat, and milk were also assessed by using the established method. The result showed a higher content of 1045.0 ± 15.6 µg/g for TMAO in fish meat and less in egg yolk (3.3 ± 0.4 µg/g), but few in the pork or milk samples. Betaine was found in four kinds of food with concentrations of 11.5 ± 1.5, 32.2 ± 3.2, 57.5 ± 2.8, and 4.2 ± 0.8 µg/g, respectively (Figure 5).

## 3. Materials and Methods

### 3.1. Chemicals and Materials

TMAO (purity ≥ 98%) and betaine (purity ≥ 98%) were purchased from Shanghai Yuanye Biological Technology Co., Ltd. (Shanghai, China). Deuterium oxide (D_2_O, 99.9% atom %D) and 3-(trimethylsilyl)-propionic-2,2,3,3-*d*_4_ acid sodium salt (TSP-*d*_4_, 98% atom %D) were purchased from Cambridge Isotope Labortories (Cambridge, FL, USA). Methanol was purchased from Sigma-Aldrich (St. Louis, MO, USA). The egg, hairtail meat, milk, and pork samples were purchased from local supermarkets.

### 3.2. Animals

All animal procedures and testing were performed according to the national legislation and local guidelines of the Laboratory Animals Center at Tianjin University of Traditional Chinese Medicine, Tianjin, China. Six-week-old C57BL/6J male and female mice were purchased from Beijing Vital River Laboratory Animal Technology Co., Ltd. (Beijing, China). Male Wistar rats (6–8 weeks old) were purchased from National Institutes for Food and Drug Control (Beijing, China). All the animals were maintained at a controlled temperature (23 ± 2 °C) with a 12 h light/dark period and acclimated with ad libitum access to a standard chow diet and water for one week. Mice were randomly divided into control and TMAO-fed groups, including a male control group (M-C, i.g. saline solution, *n* = 5), male model group (M-M, i.g. 0.2 mg/mL TMAO in water, *n* = 5), female control group (F-C, i.g. saline solution, *n* = 3), and female model group (F-M, i.g. 0.2 mg/mL TMAO in water, *n* = 3). The dosage of TMAO was 2 mg/kg per day. The intervention lasted for 7 days. Rats were randomly divided into two groups, eight rats per group, including the control group (C) that received water and model group (M) that received 15% fructose in water (W/V) ad libitum. The feeding cycle lasted for 8 weeks. At the end of these experiments, animals were fasted overnight before being euthanized. Blood samples were collected and then placed on ice in the absence of anticoagulant for 20 min before centrifugation at 4 °C, 5000× *g* rpm for 10 min. Serum samples were snap-frozen in liquid nitrogen and then stored at −80 °C for analysis.

### 3.3. Sample Preparation

HyperSep^TM^ Retain CX SPE column (Thermo Fisher Scientific Inc., Waltham, MA, USA) was preconditioned by 2 mL of methanol and then 2 mL of 2% formic acid in water. Accurately weighed TMAO and betaine were dissolved in 1 mL of water to yield standard solutions. Food sample (100 mg) was added with 800 μL of precooled methanol/water (2:1) and homogenized on ice. The supernatant was obtained after centrifugation at 4 °C, 16,000× *g* for 10 min. The standard solutions/serum/supernatant of food samples were subjected to SPE column and successively eluted with 2% formic acid in water, methanol and 5% ammonium hydroxide in methanol. Each gradient eluted in a volume of 4 mL, and the eluent was collected separately. The eluent of 5% ammonia water in methanol was dried under nitrogen, and then redissolved in 500 μL D_2_O containing 0.03 mM TSP-*d*_4_.

### 3.4. NMR Spectroscopy

^1^H-NMR spectra were recorded at 298 K on a Bruker Avance III 600 MHz spectrometer, operating at 600.13 MHz for proton, equipped with a cryogenic probe (Bruker, Biospin, Germany). Noesygppr1d pulse train was adopted and scanned 32 times (NS) with pulse width (P1) of 9.23 μs, empty sweeps of four times (DS), acquisition time (AQ) of 3.41 s, a relaxation delay (D1) of 13 s, and spectral width (SWH) of 9615 Hz. NMR original data was processed by MestReNova 6.1.0 software (Mestrelab Reaearch S.L, Spain). The inversion recovery pulse sequence was employed to determine T1 of TMAO and betaine. After phase correction and baseline correction, the ^1^H chemical shift was relative to sodium 3-(trimethylsilyl) propionate-2,2,3,3-*d*_4_ (TSP-*d*_4_) as an internal standard.

### 3.5. Statistical Analysis

Differential analysis using a two-tailed unpaired Student’s t-test was conducted using GraphPad Prism 6.0 (GraphPad Software, San Diego, CA, USA).

## 4. Conclusions

In this study, we established a ^1^H NMR-based method to quantify TMAO and betaine in biological and food samples. Due to the limited sensitivity and resolution of NMR, it is challenging to measure TMAO, betaine, and various endogenous metabolites in one spectrum without any overlap. We first performed a 1D-NOESY pulse sequence to suppress solvent signal to enhance resonance sensitivity of metabolites. Secondly, the NMR sensitivity and resolution were further enhanced by using a solid-phase extraction (SPE) procedure before sample analysis. Cation-exchange sorbent selectively absorbed alkaline metabolites, such as TMAO, betaine, and choline, and successfully removed the interferences of glucose and taurine, signals of which were severely overlapped with those of TMAO and betaine. The established method was validated with a good linearity, precision, repeatability, stability, and accuracy. The LOD and LOQ values for TMAO and betaine were much lower than the levels in normal human serum, suggesting the developed ^1^H NMR method had prospect in clinic. We also successfully estimated TMAO and betaine contents in different kinds of food, including fish meat, egg yolk, milk, and pork. These results indicated the SPE column effectively minimized the peak overlap between TMAO, betaine, and glucose, taurine, which in combination with ^1^H NMR analysis has potential as a fast and reliable method to detect TMAO and betaine in biological and food samples.

## Figures and Tables

**Figure 1 molecules-26-01334-f001:**
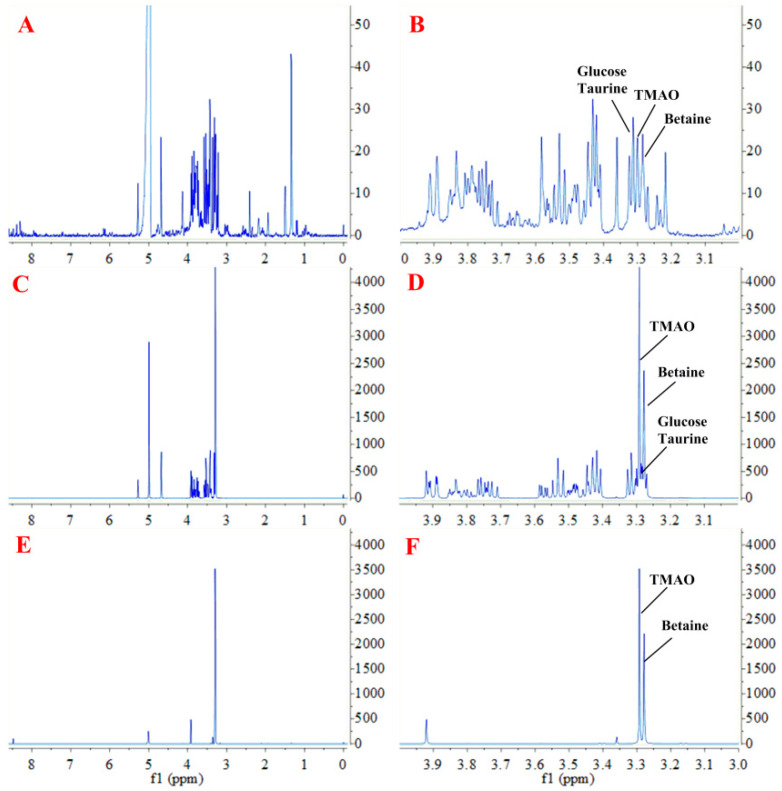
Representative proton nuclear magnetic resonance (^1^H NMR) spectra (600 MHz) of trimethylamine *N*-oxide (TMAO) and betaine: (**A**), Spectra of serum sample; (**B**), Enlarged view of A; (**C**), Spectra of standard sample not treated with solid-phase extraction (SPE); (**D**), Enlarged view of C; (**E**), Spectra of standard sample treated with SPE; and (**F**), Enlarged view of C.

**Figure 2 molecules-26-01334-f002:**
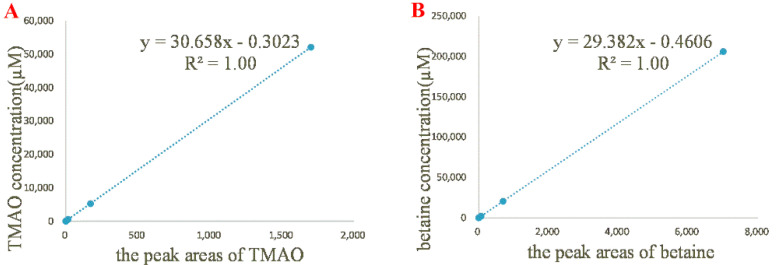
Results of linearity tests. (**A**), The calibration curve of TMAO (5.21 µM, 52.1 µM, 521 µM, 5.21 mM, and 52.1 mM); (**B**), the calibration curve of betaine (20.60 µM, 206 µM, 2.06 mM, 20.6 mM, and 206.0 mM).

**Figure 3 molecules-26-01334-f003:**
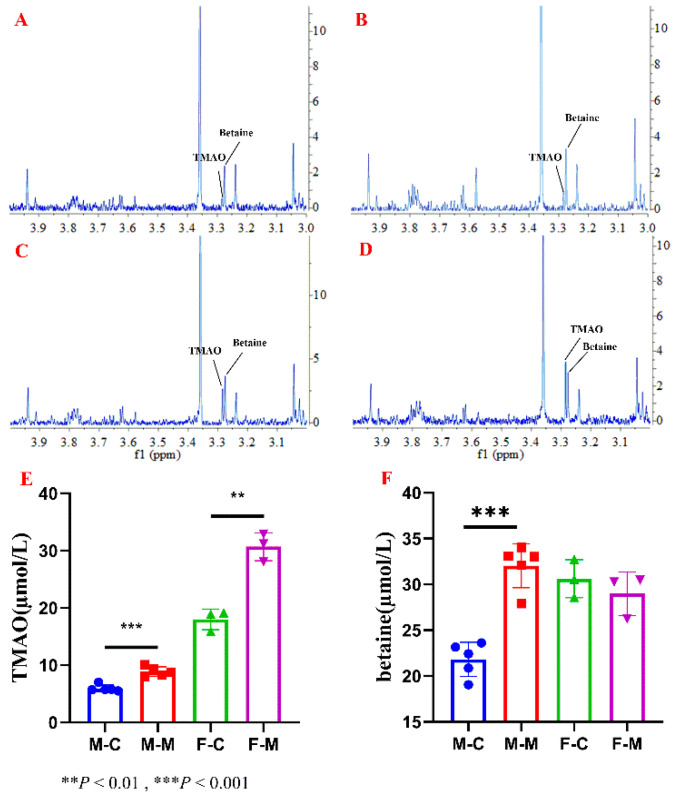
Representative ^1^H NMR spectra of SPE-treated serum sample of mouse from four groups, including (**A**) male control group, (**B**) male TMAO-fed group, (**C**) female control group and (**D**) female TMAO-fed group, (**E**) TMAO concentrations in serum samples of mouse, and (**F**) betaine concentrations in serum samples of mouse.

**Figure 4 molecules-26-01334-f004:**
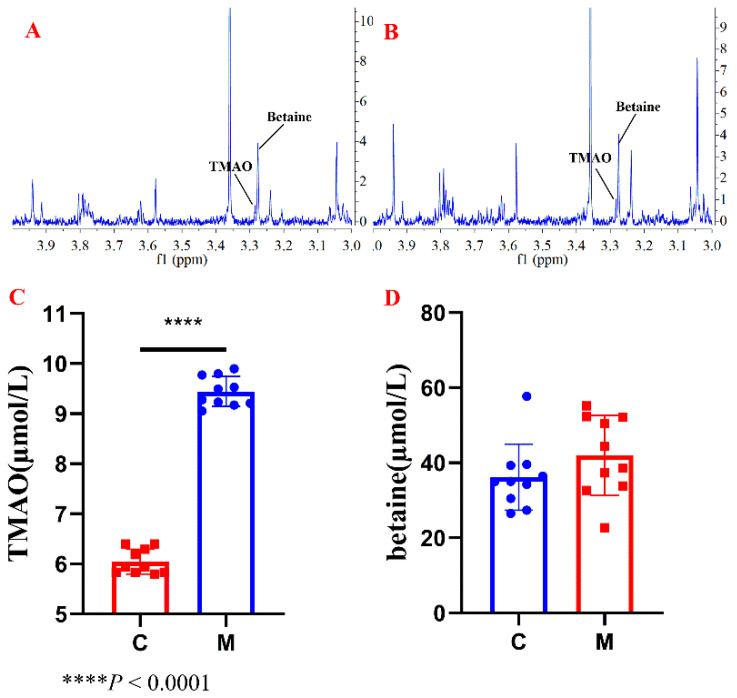
Representative ^1^H NMR spectra of SPE-treated serum sample of rats from four groups, including (**A**) control group, (**B**) fructose-fed group, (**C**) TMAO concentrations in serum samples of rats, and (**D**) betaine concentrations in serum samples of rats.

**Figure 5 molecules-26-01334-f005:**
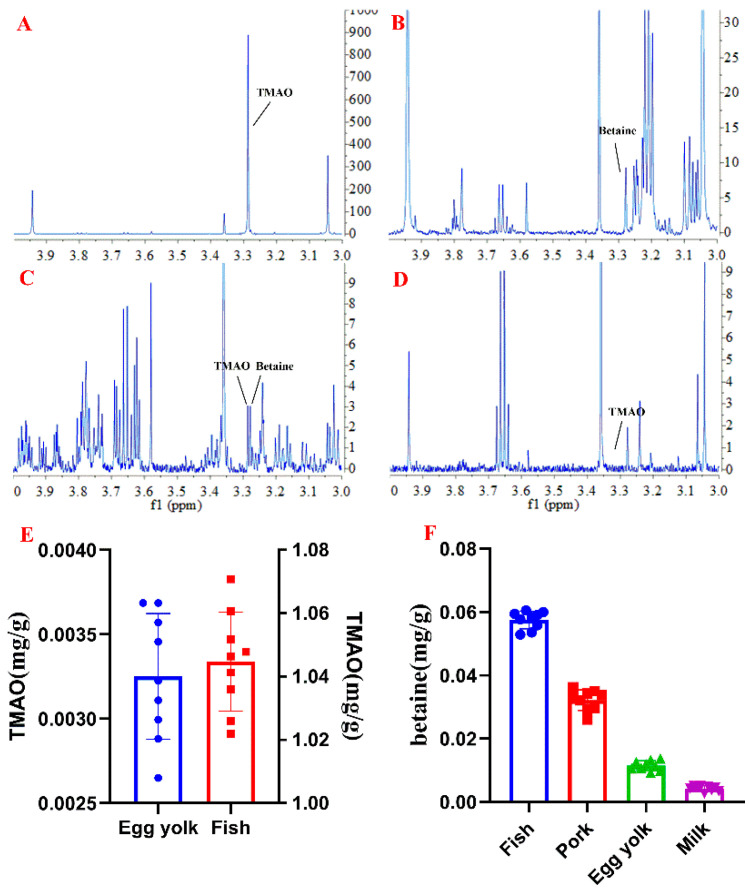
Representative ^1^H NMR spectra of SPE-treated food samples, including (**A**) fish, (**B**) pork, (**C**) egg yolk, (**D**) milk, (**E**) TMAO concentrations in food samples, and (**F**) betaine concentrations in food samples.

**Table 1 molecules-26-01334-t001:** Method validation of quantitative ^1^H NMR analysis.

	TMAO	Betaine
Chemical shift *δ* (ppm)	3.29	3.28
Linear regression equation	*y* = 30.66 *x* − 0.3023	*y* = 29.38 *x* − 0.4606
correlation coefficient	*R*^2^ = 1.00	*R*^2^ = 1.00
LOD (µM)	1.00	1.52
LOQ (µM)	3.02	4.60
Precision RSD (%)	0.85	0.93
Repeatability RSD (%)	3.06	2.76
Stability RSD (%)	2.79	2.81

**Table 2 molecules-26-01334-t002:** Spiked recovery of TMAO and betaine in standard solution.

Compound	Original (µmol)	Spiked (µmol)	Found (µmol)	Recovery (%)	Average Recovery (%) ± SD	RSD (%)
TMAO	12.11	6.21	18.21	98.23	98.48 ± 0.23	0.23
	12.06	12.14	24.02	98.52		
	12.08	18.17	30.01	98.68		
betaine	48.24	24.13	71.79	97.60	98.84 ± 0.35	0.36
	48.17	48.22	95.54	98.24		
	48.21	72.09	118.62	97.67		

**Table 3 molecules-26-01334-t003:** Spiked recovery of TMAO and betaine in serum sample.

Compound	Original (µmol)	Spiked (µmol)	Found (µmol)	Recovery (%)	Average Recovery (%) ± SD	RSD (%)
TMAO	9.53	9.87	18.21	98.12	98.47 ± 0.50	0.51
	9.52	9.87	24.02	99.04		
	9.55	9.87	30.01	98.25		
betaine	37.33	38.79	71.79	98.18	97.95 ± 0.64	0.65
	37.36	38.79	95.54	97.23		
	37.35	38.79	118.62	98.44		

## Data Availability

Not applicable.

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
