# Peer review of "Determination of Trimethylamine N-oxide and Betaine in Serum and Food by Targeted Metabonomics"

_molecules, 2021, doi:10.3390/molecules26051334_

Round 1

Reviewer 1 Report

The conclusion should be better written as it is very short. The figures show the results obtained. The methods are well described in particular the sample preparation and the pulses used. I believe it may be in my humble opinion it can be published after minor revision.

Reviewer 2 Report

The present paper described a method for targeted profiling of TMAO and betaine in serum and food samples based on a combination of one-step sample pretreatment and 1H NMR spectroscopy.  Proton signals at δ 3.29 and δ 3.28 were employed to quantify TMAO and betaine, respectively.  The developed method was examined with acceptable linear relationship, precision, stability, repeatability and accuracy.

The following points should be considered for publication:

Line 82, 194: In measurements of T1, a number of the repeated experiments should be described in method section.  Was the inversion recovery pulse sequence employed? 

Line 87: The calibration curves should be shown in supporting information to indicate the accuracy.

Line 116: In Table 1, correlation coefficient was provided as “R2 = 1”.  Three effective digits should be shown.

Line 195: The temperature was 298 K in the NMR measurements.  This reviewer wonders that another temperature might have provided much better dispersion of two 1H signals, closely resonated at δ 3.29 and δ 3.28.  Some evidence which indicated that the temperature of 298 K was the most appropriate one, should be described.

Line 201: Regarding the description “the proton signal of TSP-d4 was calibrated as δ 0 ppm”, it should be “the 1H chemical shift was relative to sodium 3-(trimethylsilyl)propionate-2,2,3,3-d4 (TSP-d4) as an internal standard.”  

Reviewer 3 Report

Regardless the work of He et al. sounds interesting for application, it is not clear to me that the authors claim “One of the key points of this study was the preparation of biological and food samples”, while no absolute recovery tests seems to be done. So, instead of the estimation of accuracy for 1H NMR measurements, an absolute recovery test, retrieving standards added to the biologicals sample (and subjected to all steps of the extraction/purification procedure) should be added.

Round 2

Reviewer 3 Report

The authors answered in part to my questions and I suggest that they complete the work with recovery tests performed on serum or food sample.

Author Response

This manuscript is a resubmission of an earlier submission. The following is a list of the peer review reports and author responses from that submission.